# Clinico-Radiological Outcomes in WNT-Subgroup Medulloblastoma

**DOI:** 10.3390/diagnostics14040358

**Published:** 2024-02-07

**Authors:** Shakthivel Mani, Abhishek Chatterjee, Archya Dasgupta, Neelam Shirsat, Akash Pawar, Sridhar Epari, Ayushi Sahay, Arpita Sahu, Aliasgar Moiyadi, Maya Prasad, Girish Chinnaswamy, Tejpal Gupta

**Affiliations:** 1Department of Radiation Oncology, ACTREC/TMH, Tata Memorial Centre, Homi Bhabha National Institute, Kharghar, Navi Mumbai 410210, India; msvshakthi1994@gmail.com (S.M.); chatterji08@gmail.com (A.C.); archya1010@gmail.com (A.D.); 2Neuro-Oncology Laboratory, ACTREC/TMH, Tata Memorial Centre, Homi Bhabha National Institute, Kharghar, Navi Mumbai 410210, India; neelamshirsat@hotmail.com; 3Clinical Research Secretariat, ACTREC/TMH, Tata Memorial Centre, Homi Bhabha National Institute, Kharghar, Navi Mumbai 410210, India; akash.bstats@gmail.com; 4Department of Pathology, ACTREC/TMH, Tata Memorial Centre, Homi Bhabha National Institute, Kharghar, Navi Mumbai 410210, India; sridhep@gmail.com (S.E.); ayujain24@gmail.com (A.S.); 5Department of Radio-Diagnosis, ACTREC/TMH, Tata Memorial Centre, Homi Bhabha National Institute, Kharghar, Navi Mumbai 410210, India; drarpitasahu@gmail.com; 6Department of Neurosurgery, ACTREC/TMH, Tata Memorial Centre, Homi Bhabha National Institute, Kharghar, Navi Mumbai 410210, India; aliasgar.moiyadi@gmail.com; 7Department of Pediatric Oncology, ACTREC/TMH, Tata Memorial Centre, Homi Bhabha National Institute, Kharghar, Navi Mumbai 410210, India; maya.prasad@gmail.com (M.P.); girish.c.tmh@gmail.com (G.C.)

**Keywords:** medulloblastoma, multimodality therapy, relapse, survival, WNT

## Abstract

Medulloblastoma (MB) comprises four broad molecular subgroups, namely wingless (WNT), sonic hedgehog (SHH), Group 3, and Group 4, respectively, with subgroup-specific developmental origins, unique genetic profiles, distinct clinico-demographic characteristics, and diverse clinical outcomes. This is a retrospective audit of clinical outcomes in molecularly confirmed WNT-MB patients treated with maximal safe resection followed by postoperative standard-of-care risk-stratified adjuvant radio(chemo)therapy at a tertiary-care comprehensive cancer centre. Of the 74 WNT-MB patients registered in a neuro-oncology unit between 2004 to 2020, 7 patients accrued on a prospective clinical trial of treatment deintensification were excluded, leaving 67 patients that constitute the present study cohort. The median age at presentation was 12 years, with a male preponderance (2:1). The survival analysis was restricted to 61 patients and excluded 6 patients (1 postoperative mortality plus 5 without adequate details of treatment or outcomes). At a median follow-up of 72 months, Kaplan–Meier estimates of 5-year progression-free survival and overall survival were 87.7% and 91.2%, respectively. Traditional high-risk features, large residual tumour (≥1.5 cm^2^), and leptomeningeal metastases (M+) did not significantly impact upon survival in this molecularly characterized WNT-MB cohort treated with risk-stratified contemporary multimodality therapy. The lack of a prognostic impact of conventional high-risk features suggests the need for refined risk stratification and potential deintensification of therapy.

## 1. Introduction

Medulloblastoma (MB), the most common malignant tumour involving the brain and central nervous system (CNS) in children is now considered to be a heterogeneous disease composed of four broad molecular subgroups, namely wingless (WNT), sonic hedgehog (SHH), Group 3, and Group 4, respectively (Table 1), with subgroup-specific developmental origins, unique genetic profiles, distinct clinico-demographic characteristics, and diverse clinical outcomes [1,2,3]. This led to the incorporation of molecular/genetic information in the World Health Organization’s (WHO) CNS tumour classification that recommended an integrated layered diagnosis including available molecular/genetic information in the WHO 2016 update [4] and the 5th edition of the WHO classification of tumours involving the CNS (WHO CNS 5) [5].

MB has a high propensity to spread throughout the craniospinal axis via cerebrospinal fluid (CSF) pathways, with metastatic disease being identified on neuraxial staging in nearly one-third of patients at initial diagnosis [2,3] necessitating treatment of the entire brain and spinal cord, including its covering meninges for disease control. Recently, the existence of circulating tumour cells (CTCs) in the peripheral blood of patients with MB has been demonstrated [6] and explains the rare phenomenon of extraneural metastases (ENM) via the hematogenous route seen in <1% of patients at initial diagnosis. Occasionally, these CTCs can migrate towards the leptomeninges, leading to neuraxial dissemination [6]. A lymphatic network has also been described in the CNS, particularly in the meninges (within the dura mater), which facilitates CSF drainage, part of which is in the subarachnoid space and drains into the cervical lymph nodes that connect with the lymphatic circulation, thereby incriminating CNS lymphatics as a potential pathway of spread [6]. Contemporary management for noninfantile MB [2,3,7] comprises maximal safe resection followed by postoperative risk-stratified adjuvant craniospinal irradiation (CSI) to a dose of 23.4–36 Gy/13–20 fractions plus boost irradiation of the tumour-bed to a dose of 18–30.6 Gy/10–17 fractions, resulting in a total primary site radiotherapy (RT) dose of 54 Gy/30 fractions. This is followed by 6–9 cycles of multiagent adjuvant systemic chemotherapy [2,3,8]. Traditionally, children over the age of 3 years at diagnosis with no/small residual tumour (<1.5 cm^2^) and an absence of metastatic disease (M0) were classified as average risk for disease [9], with >80% long-term survival [10,11,12], while younger age (<3 years), large residual tumour (≥1.5 cm^2^), and presence of leptomeningeal metastases (M+ disease) either alone or in combination were considered high-risk features [9], yielding much worse 5-year survival (30–60%) despite aggressive multimodality therapy [12,13]. This traditional risk stratification has been further refined by incorporating molecular/genetic information in the contemporary molecular era into low-risk, standard-risk, high-risk, and very high-risk categories with an expected 5-year overall survival of >90%, 75–90%, 50–75%, and <50%, respectively [14].

Intensive multimodality treatment achieves good survival outcomes in MB but is associated with significant acute and late treatment-related toxicities. Aggressive surgical resection can be associated with increased postoperative complications, such as cerebellar mutism syndrome, which may evolve into persistent cognitive dysfunction, speech deficits, and ataxia. Determining the optimal balance between the extent of resection to improve prognosis while respecting the surrounding critical neural structures to minimize morbidity has been challenging for the neurosurgical community. The prognostic value of MB extent of resection gets attenuated after accounting for molecular subgroup affiliation [15]; however, such information is typically not available prior to surgery. Radiogenomics, i.e., the extraction of semantic and/or radiomic features from preoperative magnetic resonance imaging (MRI) to predict the molecular subgrouping of MB [16], though gaining popularity, has too suboptimal a diagnostic accuracy presently to be used in clinical practice. Adjuvant radio(chemo)therapy in MB is associated with substantial morbidity, including but not limited to neuro-cognitive impairment; neuro-psychological dysfunction; endocrinopathy, particularly growth retardation; sensori-neural hearing loss (SNHL); vasculopathy specially cerebro-vascular accident (CVA); and second malignant neoplasm (SMN) [17,18]. Of all MBs, the WNT subgroup has the best outcomes (5-year survival > 90%), particularly in children [2,3,12], making these long-term survivors more susceptible to dose-dependent treatment-related late morbidity prompting systematic attempts at the deintensification of therapy [19]. An appropriate risk-classification schema and optimal treatment regimen for WNT-MB is yet to be defined. For this, it is important to identify prognostic factors and assess patterns of failure to guide therapeutic decision-making for tailoring adjuvant therapy in the WNT subgroup MB.

## 2. Materials and Methods

Patients with molecularly confirmed WNT-MB treated with maximal safe resection followed by postoperative standard-of-care risk-stratified adjuvant radio(chemo)therapy were identified retrospectively via an electronic search of the neuro-oncology database.

*Molecular subgrouping*: the molecular subgroup assignment of MB was based on an inhouse-developed assay combining differential expression of 12 protein-coding genes and 9 miRNAs using real-time reverse transcriptase polymerase chain reaction (RT-PCR), as described previously [20]. Briefly, RNA (1–2 mg) was reverse transcribed using random hexameric primers and M-MLV reverse transcriptase (Invitrogen, Thermo Fisher Scientific Inc. Waltham, MA, USA). The primers for real-time PCR analysis were designed such that they corresponded to 2 adjacent exons and, wherever possible, were located at exon boundaries to avoid the amplification of genomic DNA. The amplicon size was maintained below 75–80 bp, so as to enable amplification of the fragmented RNA from formalin-fixed paraffin-embedded (FFPE) tissues. The expression was analysed by a SYBR Green PCR amplification assay on an Applied Biosystems 7900HT real-time PCR system using 10 ng cDNA per reaction for frozen tissues and 10–100 ng cDNA per reaction for FFPE tissues. For miRNA expression analysis, 50 ng RNA from fresh tissues and 50–200 ng RNA from FFPE tissues were reverse transcribed using multiplex RT primer pools and the Taqman MicroRNA Reverse Transcription Kit (Applied Biosystems, Thermo Fisher Scientific Inc. Waltham, MA, USA) according to the manufacturer’s instructions. The expression of each miRNA was analysed by a TaqMan real-time miRNA assay (Applied Biosystems) on the ABI 7900HT real-time PCR system using 10 ng cDNA from frozen tissues and 10–40 ng cDNA from FFPE tissues. The relative quantity (RQ) of each protein-coding gene/miRNA compared with GAPDH/RNU48 was determined by the comparative cycle threshold (Ct) method. Genes that were significantly differentially expressed in the 4 molecular subgroups were identified by a Significance Analysis of Microarray (MeV, http://www.TM4.com, (accessed on 3 March 2013)) of expression profiling data previously obtained using Affymetrix Gene 1.0 ST array. The selection of 12 protein-coding marker genes for classification from the significantly differentially expressed genes was based on the standardized fold-change in the expression of the gene in a particular subgroup. Concomitant overexpression of *WIF1*, *DKK2*, and *MYC* identified WNT-MB. Overexpression of *HHIP*, *EYA1*, and *MYCN* with underexpression of *OTX2* served as markers for the SHH-subgroup. The overexpression of *EOMES* helped to identify Group 3 and Group 4 tumours, while a higher expression of *NPR3*, *MYC*, and *IMPG2* with a lower expression of *GRM8* and *UNC5D* helped to distinguish Group 3 from Group 4 tumours. Similar to gene-expression profiling, the differential expression of 9 selected miRNAs was used for subgroup assignment. WNT-activated tumours showed significant overexpression of miR-193a-3p, miR-224, miR-148a, miR-23b, and miR-365 compared with other subgroups. MiR-182 was found to be overexpressed in all WNT-MBs and in many Group 3 and some Group 4 MBs, while miR-204 was overexpressed in all WNT-MBs and in most Group 4 MBs. MiR-182, miR-135b, and miR-204 were found to be underexpressed in SHH-activated MBs. MiR-135b was found to be overexpressed in Group 3 and Group 4 tumours. MiR-592, a miRNA that is located within the *GRM8* gene, was overexpressed in Group 4 MB. This aforesaid assay had previously been successfully validated [20] against a set of 34 well-annotated FFPE MB samples with subgroup assignment based on the 22-gene set NanoString assay from the German Cancer Research Centre (DFKZ). In recent times (after 2017), confirmation of WNT activation was further supplemented by testing for monosomy 6 (fluorescence in situ hybridization), *CTNNB1* mutation analysis (Sanger sequencing), and/or nuclear beta-catenin positivity (immunohistochemical staining) as orthogonal techniques.

*Treatment and follow-up*: information regarding patient demographics, clinical features, histopathological features, molecular profiling, risk stratification, treatment details, and outcomes were retrieved from hospital case files and/or electronic medical records as appropriate. All patients underwent maximal safe resection followed by postoperative risk-stratified adjuvant radio(chemo)therapy. Risk stratification after surgery was based on conventional criteria without upfront knowledge of the molecular subgroup. Children (≤16 years) with average-risk MB defined as residual tumour < 1.5 cm^2^ with no evidence of metastases (M0) were treated with CSI (23.4 Gy) plus boost irradiation (30.6 Gy) for a total primary-site dose of 54 Gy followed by 6 cycles of adjuvant systemic chemotherapy. For adolescents and young adults (AYA) over 16 years of age at initial diagnosis with average-risk MB, RT alone was considered and comprised full-dose CSI (35–36 Gy) plus boost (18–19.8 Gy) for a total primary-site dose of 54–54.8 Gy without adjuvant chemotherapy. The presence of any high-risk features, such as large residue (≥1.5 cm^2^), metastatic disease (M+), or adverse histology such as large-cell or anaplastic (LC/A) mandated full-dose/extended-dose CSI (35–40 Gy) plus boost irradiation of the primary site (14.4–19.8 Gy) with or without boost (5.4–9 Gy) to the metastatic deposits followed by 6 cycles of adjuvant systemic chemotherapy. Following completion of adjuvant radio(chemo)therapy, patients were followed up clinically at 3–4 monthly intervals for the first two years, 6 monthly intervals for 5 years, and annually thereafter with periodic surveillance MRI scans as per institutional policy.

*Statistical analysis*: clinical and demographic variables were analysed and summarized using descriptive statistics with measures of central tendency and dispersion being reported. Patterns of relapse were defined as local recurrence (in and around the surgical cavity/resected tumour bed); metastatic disease either involving the leptomeningeal space outside the initial tumour bed in the cranial and/or spinal leptomeninges or ENM involving the bones, lymph nodes, or bone marrow; or a combination of the above. Progression-free survival (PFS) was defined as the time interval from the date of surgery till documented clinico-radiological progression, death due to any cause, or last follow-up. Overall survival (OS) was defined from the date of surgery till death due to any cause or last documented follow-up. The median follow-up of surviving patients was calculated by the reverse Kaplan–Meier method. Time-to-event outcomes were analysed using the product-limit method of Kaplan–Meier and presented as 5-year estimates with a 95% confidence interval (CI). Univariate analysis of variables of known and/or presumed prognostic significance was done using the log-rank test after dichotomization at median values or cutoffs established from earlier literature as appropriate. Statistical analysis was performed using SPSS version 25.0 (IBM Corporation, Armonk, NY, USA) and R Studio version 3.2.7 (R Corporation, Vienna, Austria). The study was duly reviewed and approved by the Institutional Ethics Committee (IEC) which functions in accordance with the Declaration of Helsinki. IEC also granted a waiver of consent due to the retrospective nature of the study with no/minimal risk to participants.

## 3. Results

An electronic search of the neuro-oncology database identified a total of 504 MB patients registered in the neuro-oncology unit of the institute between 2004 and 2020, of which 74 (14.6%) were diagnosed as having WNT-subgroup MB [20]. Seven patients who were treated on a prospective protocol of therapy deintensification in WNT-MB [21] were excluded from the dataset, leaving 67 patients who constitute the present study cohort.

*Clinico-demographic features*: patient, disease, and treatment characteristics of the study cohort are summarized in Table 2. The median age of the study cohort was 12 years with an interquartile range (IQR) of 9–18 years and a preponderance of male gender (2:1 ratio). Pediatric WNT-MB (defined as age ≤ 16 years) comprised 73.1% (*n* = 49) of patients compared to 26.9% (*n* = 18) of AYA WNT-MB (defined as age > 16 years). All patients underwent maximal safe resection with gross total resection (GTR) achieved in 49% of patients. Classic histology was the most common histological subtype seen in 61.2% (*n* = 41) of patients. Metastatic disease status by CSF cytology and/or neuro-imaging was available in 62 patients with the majority (91.9%, *n* = 57) being nonmetastatic at initial diagnosis. The presence of any one or more of the following adverse features, such as large residual tumour (≥1.5 cm^2^), metastatic disease (M+), and LC/A histology classified 16 (29.1%) patients as having high-risk disease and 39 (70.9%) patients as average-risk disease. All included patients were treated postoperatively with contemporary risk-stratified RT comprising CSI plus boost irradiation with or without adjuvant systemic chemotherapy. The median dose of CSI was 35 Gy (IQR: 23.4–35 Gy) with a median tumour-bed boost dose of 19.8 Gy (IQR: 19.8–30.6 Gy). Extended dose CSI (40 Gy) and boost irradiation of metastatic deposits were also done at the discretion of the treating radiation oncologist. Most of the patients were treated with conformal techniques either three-dimensional conformal radiotherapy (3D-CRT) or intensity-modulated radiation therapy (IMRT) using six MV photons on modern linear accelerators, including tomotherapy. Adjuvant systemic chemotherapy was delivered in 72.2% (*n* = 39) of patients, whereas 27.8% (*n* = 15) of patients did not receive any chemotherapy after the completion of RT. Chemotherapy was initiated 4–6 weeks after the completion of RT after sufficient myelo-recovery, defined as absolute neutrophil count (ANC) >1500/dl and platelet count > 100,000/dL. Adjuvant chemotherapy generally comprised six cycles of cisplatin (75 mg/m^2^ intravenously on d1 in alternate cycles two, four, and six), cyclophosphamide (1000 mg/m^2^ intravenously on d1–d2 in cycles one, three, and five and d2–d3 in cycles two, four, and six) and vincristine (1.5 mg/m^2^ intravenously d1 and d8 in all six cycles) given at 4-week intervals with adequate hydration, forced saline diuresis, mesna prophylaxis, and requisite dose modifications as appropriate [7]. Two of five children with metastatic disease at initial diagnosis also received 1 year of maintenance chemotherapy postcompletion of standard therapy using the modified combined oral metronomic biodifferentiating antiangiogenic therapy (COMBAT) regimen comprising temozolomide, etoposide, celecoxib, fenofibrate, and retinoic acid.

*Patterns of failure, causes of death, and survival outcomes*: 6 patients (1 postoperative mortality and 5 without adequate details of treatment or outcomes) were excluded from the survival analysis which was restricted to 61 patients. Nine of the 61 included patients who experienced an event of interest (relapse and/or death). Eight patients were detected with relapse on follow-up with leptomeningeal dissemination seen in 5 patients (including two with synchronous local recurrence), local tumour-bed recurrence in 4 patients (including 2 with synchronous neuraxial relapse), and isolated extra-neural metastases (lymph nodes, bones) in a single patient. Images from one such case scenario each of tumour-bed recurrence only, synchronous local recurrence with neuraxial failure, and isolated ENM from the study cohort are illustrated in Figure 1. Seven of 61 patients died by the time of this analysis, 6 of recurrent/progressive disease and 1 due to chemotherapy-induced febrile neutropenia leading to septic shock and death. Clinico-demographic details, pattern of relapse, and outcomes of all these nine patients experiencing an event are summarized in Table 3. Of the five WNT-MB patients who were treated with salvage therapy at relapse, two patients (W1: tumour-bed recurrence and W2: diffuse leptomeningeal metastases) achieved postrelapse survival of 55 and 29 months, respectively, while the lone patient with ENM (W8) was alive with disease on salvage systemic chemotherapy at the time of this analysis. Two patients treated with re-excision alone without further reirradiation or salvage chemotherapy succumbed to further progressive disease within 6–9 months of relapse. All three WNT-MB patients offered the best supportive care at relapse and died of progressive disease within 3 months of the first relapse. At a median follow-up of 72 months (IQR: 51–101 months) for the entire study cohort (*N* = 61), the 5-year Kaplan–Meier estimates of PFS and OS were 87.7% (95%CI: 75.1–96.1%) and 91.2% (95%CI: 83.0–100%), respectively (Figure 2). Univariate analysis of various patient-, disease-, and treatment-related factors did not identify any putative prognostic factor impacting upon PFS or OS (Table 4). A multivariate analysis was considered inappropriate due to the small number of events in the study cohort.

## 4. Discussion

The clinico-demographic characteristics of this large cohort of WNT-MB patients treated at an academic neuro-oncology unit of a tertiary care comprehensive cancer centre are largely in accordance with the previously published literature with minor differences. The present study had more males with WNT-MB than females (2:1), possibly due to socio-cultural differences and the patriarchal mindset prevalent in the low-middle-income country setting in Southeast Asia compared to the fairly balanced gender ratio reported previously from high-income countries of the West [22]. The median age at diagnosis of the present study cohort was also slightly higher (12 years, IQR: 9–18 years) reflecting an increased representation of adult WNT-MB compared to an international reference cohort (median 10 years, IQR: 8–14.2 years) which was largely limited to the pediatric age group [22].

Given the low prevalence of WNT-MB (constituting around 10% of all MBs) coupled with a very low risk of failure in appropriately treated patients, prognostic factors impacting upon survival, patterns of relapse, and drivers of metastatic dissemination are relatively poorly understood. Nobre et al. [22] assembled a retrospective multi-institutional clinically annotated cohort of 93 WNT-pathway medulloblastoma patients using an integrated genomic approach. Fifteen patients with relapse were identified, 12 in the metastatic compartment, including 1 with ENM and 3 in the surgical cavity. Interestingly, 8 of 11 neuraxial relapses were in lateral ventricles (6 confined to frontal horns), leading to the hypothesis that the unique microenvironment of the ependymal lining of the lateral ventricles may be more conducive to the homing of WNT-MB. Maintenance systemic chemotherapy (*p* = 0.033), specifically a lower cumulative dose of cyclophosphamide/ifosfamide (<12 mg/m^2^), was reported to be associated with an increased risk of relapse. It was proposed that the paracrine signals driven by mutant β-catenin protein induce a fenestrated tumour vasculature promoting the accumulation of chemotherapeutic agents within the tumour bed. The authors also reported that male gender (*p* = 0.032) was associated with a significantly increased risk of relapse in WNT-MB. Age at diagnosis, extent of resection, metastatic status at presentation, dose of CSI, and additional molecular/genetic alterations did not predict the risk of relapse in their study. In another cohort of 191 WNT-MB patients registered in the HIT database [23], mutations in *CTNNB1*, *APC*, and *TP53* were analysed by DNA sequencing and chromosomal copy number aberrations by molecular inversion probe technology to identify the prognostic impact of *TP53* mutations and other chromosomal aberrations in the WNT subgroup. Patients with tumours harbouring the *TP53* mutation showed worse outcomes (5-year PFS: 68% vs. 93%, *p* = 0.001 and 5-year OS: 81% vs. 95%, *p* = 0.105) compared to *TP53* wild-type tumours. Gain of OTX2 was associated with inferior survival outcomes (5-year PFS: 72% vs. 93%, *p* = 0.017 and 5-year OS: 83% vs. 97%, *p* = 0.006). A multivariable Cox regression analysis identified both genetic alterations as independent prognostic markers for survival, raising concerns regarding the inclusion of such patients in ongoing prospective trials of therapy deintensification.

The presence of intratumoural heterogeneity within the four broad molecular subgroups prompted several researchers to perform large-scale integrative clustering analysis combining DNA methylation and gene-expression profiling to identify further subtypes within each broad molecular subgroup [24,25,26] resulting in a consensus definition of 12 subtypes of MB in second-generation molecular subgrouping [27]. WNT-pathway MB typically demonstrates homogenous genome-wide expression patterns and methylation profiles; however, two molecular subtypes of WNT-activated MB have been identified [27] and referred to as WNT-α and WNT-β which differ in age at diagnosis (median age of 10 vs. 20 years), frequency of monosomy 6 (>85% vs. <50%), histo-morphology (typically classic vs. sometimes LC/A), and metastatic disease (absent vs. occasionally present), respectively. Very rarely, WNT-MB may harbour the distinct genetic alterations typical of another molecular subgroup (such as SHH and non-WNT/non-SHH) in addition to WNT-activation referred to as hybrid molecular subtypes [28], indicating intratumoural heterogeneity with potential prognostic implications. *MYC* oncogenes are the most commonly amplified loci in MB [29,30] that are generally associated with non-WNT/non-SHH disease (particularly subgroup 3), LC/A histology, and metastatic dissemination, making them known biomarkers of poor prognosis. Although overexpression of MYC can also be seen in WNT-subgroup MB with no detrimental impact on survival [30], *MYC*-amplification has rarely been described [31] in WNT-activated tumours. It may be pertinent to note that increased *MYC* signalling has been shown to accelerate tumour growth and promote metastases in a murine model of WNT-MB [32].

Although MB largely remains a disease of childhood with a much lower incidence in the AYA population, it is a common perception that excellent survival outcomes achieved in the pediatric population may not be exactly mirrored in the AYA cohort [33,34]. However, contrary to popular belief, analysis of the Surveillance, Epidemiology, and End Results (SEER) database [35] from 1992–2013 reported comparable 2-year, 5-year, and 10-year survival outcomes between childhood (*n* = 616) and adult MB (*n* = 349). The first comprehensive molecular analysis of adult MB [36] defined three broad molecular subgroups, viz. WNT, SHH, and Group D (later reclassified as Group 4), with an absence of Group 3 tumours. The authors reported a worse prognosis of adult WNT-MB and Group 4 disease compared to corresponding subgroups of childhood MB; however, survival in the SHH subgroup MB was similar across both age groups. In another large multi-institutional dataset of adult MB [37], there was no prognostic impact of molecular subgrouping with a 5-year survival of 45%, 67%, 62%, and 67% for WNT, SHH, Group 3, and Group 4, respectively. The largest integrative analysis of adult MB [38] also reported no statistically significant survival differences between the four broad molecular subgroups with 5-year PFS (95%CI) of 64.4% (48.0–86.5%), 61.9% (51.6–74.2%), 80.0% (51.6–100%), and 44.9% (28.6–70.7%) for WNT (*n* = 30), SHH (*n* = 112), Group 3 (*n* = 6), and Group 4 (*n* = 41), respectively. However, what stands out clearly is the substantially lower survival in adults with WNT-activated MB (5-year survival 45–70%) compared to benchmark outcomes in childhood WNT-MB (5-year survival > 90%). However, this notion has recently been challenged with molecular subgrouping emerging as a significant prognostic factor in AYA-MB. In a large single-institutional dataset [39] of molecularly characterized AYA-MB (≥15 years at initial diagnosis), the reported 5-year survival was 87.5%, 62.2%, and 50.1% for the WNT (*n* = 14), SHH (*n* = 71), and non-WNT/non-SHH (*n* = 21) subgroups, respectively. A comparative analysis of pediatric versus AYA WNT-MB also reported similar survival outcomes [40], suggesting that age alone should not be used to intensify treatment in WNT-MB.

The time to recurrence (early vs. delayed), the pattern of failure (local, metastatic, or combined) and postrelapse survival in MB is largely dictated by disease biology and varies across the four broad molecular subgroups [41,42,43]. Time to relapse even within the WNT-subgroup has been variable across studies with both early as well as delayed relapses being reported [22,34] sometimes even beyond 10 years from initial diagnosis. Management of relapsed MB after appropriate and adequate upfront radio(chemo)therapy is not clearly defined, with no universally acceptable standard-of-care salvage treatment [44]. A substantial and large proportion of these patients, particularly with disseminated disease, are offered best supportive care alone, with only a small minority being treated with aggressive multimodality salvage therapies, including a combination of re-excision (isolated local relapse), reirradiation, and systemic therapies that might include high-dose chemotherapy with autologous stem-cell rescue and targeted therapy as appropriate. The prognosis of relapsed MB in patients previously treated with CSI in the upfront setting is typically poor and considered noncurative, with <5% long-term survival despite aggressive salvage therapies [44]. However, it is now being increasingly appreciated that postrelapse outcomes might be somewhat subgroup dependent, with WNT-MB and Group 4 tumours demonstrating a more indolent clinical course with favourable outcomes compared to SHH-MB and Group 3 disease. This study also reported favourable outcomes in a subset of relapsed WNT-MB patients, further raising the question of whether these patients can be treated with upfront deintensified therapy at initial diagnosis and reserving treatments associated with high morbidity at the time of relapse. The patterns of failure and 5-year survival outcomes of appropriately treated WNT-MB patients in various molecularly-informed prospective cohort studies, including randomized controlled trials, are summarized in Table 5 [11,12,42,45,46,47,48] which reaffirms excellent prognosis and provides justification for ongoing global efforts towards the de-escalation of therapy [19]. However, such an approach warrants caution, as two prospective deintensification studies had to be terminated prematurely due to an unacceptably high risk of failures. The first of these [21] treated rigorously defined low-risk WNT-MB patients with focal-only conformal RT to the index tumour bed (54 Gy) plus adjuvant systemic chemotherapy with the omission of upfront CSI. Three of the first seven patients accrued in the study were detected with neuraxial dissemination within 2 years of index diagnosis and, although they were subsequently successfully salvaged in the short term with aggressive multimodality therapy including full-dose CSI and systemic chemotherapy, mature outcomes of salvage therapy remain to be reported. The second study [49] utilized a postsurgery primary chemotherapy approach, eliminating RT completely in low-risk WNT-MB. Once again, three of the first six children in the study developed local recurrence and neuraxial dissemination shortly after completing chemotherapy, leading to early closure due to safety concerns. Two of them were successfully salvaged with RT including CSI plus additional chemotherapy, but one child succumbed to further progressive disease at 35 months from initial diagnosis. Of the remaining three patients, two children proceeded to immediate RT after the completion of primary chemotherapy to protect against early relapse, while the remaining child was switched midtreatment to high-dose chemotherapy with autologous stem-cell rescue. Both these studies reinforce the need for RT, particularly CSI, for effective disease control even in low-risk, favorable-biology WNT-pathway MB [50].

*Strengths and limitations*: this study represents the largest descriptive analysis of WNT-MB treated with contemporary risk-stratified radio(chemo)therapy at a single institution anywhere in the world. Access to advanced molecular diagnostics for subgroup assignment and therapeutic decision making in a multidisciplinary neuro-oncology clinic add further strength to the study. However, despite the above-mentioned strengths, several caveats and limitations remain. The retrospective nature of the study makes it susceptible to inherent biases that could potentially confound the interpretation of results. Various platforms exist for the robust molecular subgrouping of MB, including an immunohistochemistry panel, gene-expression analysis, microRNA profiling, and DNA methylation array. The study used combined gene-expression analysis and microRNA profiling for molecular subgroup assignment; however, DNA methylation, which is considered the current gold standard and method of choice for molecular classification of MB, was not performed due to issues with availability, accessibility, and affordability. Although rare, the co-occurrence of additional genetic alterations to identify any hybrid molecular subtypes was not assessed in the study. Analysis of survival outcomes was restricted to 61 patients (after excluding 6 patients) which could be a potential source of bias. Follow-up duration, though long (median of 72 months), may be considered inadequate to capture very delayed relapses and SMNs. Although therapeutic decision making was largely based on discussion in a multidisciplinary tumour board, patients may not have been treated uniformly over the long period of the study, potentially impacting upon outcomes. Finally, the lack of documented data on neuro-cognitive impairment, neuro-psychological dysfunction, SNHL, endocrinopathies, CVA, and the resultant quality of life precludes assessment of the impact of treatment-related late toxicity in these long-term survivors.

## 5. Conclusions

Medulloblastoma is a heterogeneous disease comprising four broad molecular subgroups (WNT, SHH, Group 3, and Group 4) with subgroup-specific developmental origins, unique genetic profiles, distinct clinico-demographic characteristics, and varying clinical outcomes. WNT-MB has the best survival outcomes, whereas Group 3 MB is associated with the worst prognosis. This retrospective clinical audit confirms excellent survival in WNT-MB patients treated with contemporary multimodality therapy comprising maximal safe resection followed by risk-stratified appropriate radio(chemo)therapy. The lack of prognostic impact of conventional high-risk features suggests the need for refined risk stratification and the potential for deintensification of therapy in WNT-MB.

## Figures and Tables

**Figure 1 diagnostics-14-00358-f001:**
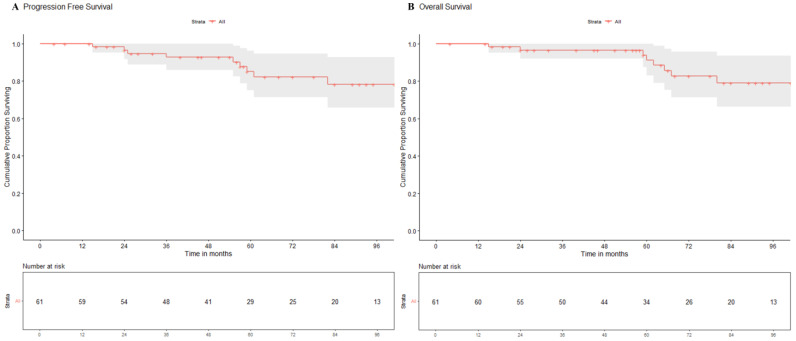
Kaplan–Meier curves of progression-free survival (**A**) and overall survival (**B**) for WNT-pathway medulloblastoma in the study cohort.

**Figure 2 diagnostics-14-00358-f002:**
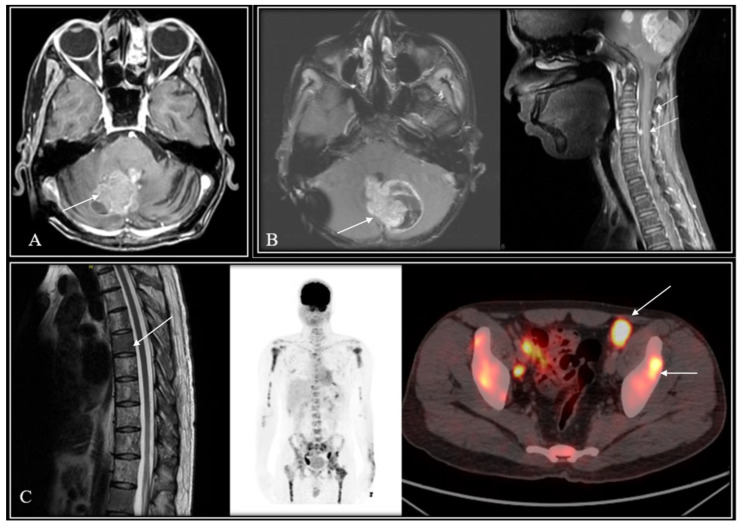
Representative images of three patients from the study cohort with local recurrence in the tumour bed (**A**), synchronous local relapse with spinal deposits (**B**), and extraneural metastases (**C**) in the WNT-subgroup medulloblastoma. White arrows highlight the site of relapse in all three cases.

**Table 1 diagnostics-14-00358-t001:** Subgroup-specific clinico-biological characteristics of medulloblastoma.

Characteristics	WNT	SHH	Group 3	Group 4
Proportion of patients	10% (Rarest)	30% (2nd common)	25% (Common)	35% (Most common)
Age at initial presentation	Older children and adolescents (rarely in infants)	All age groups (2 peaks—infants and adults)	Infants and young children (rarely in adults)	All age groups (commonly in children)
Male–Female ratio	Equal (1:1)	Equal (1:1)	Favors males (2:1)	Male preponderance (3–4:1)
Histo-morphology	Classic, rarely LC/A	ND/MBEN or Classic	LC/A, Classic	Classic, LC/A
Leptomeningeal metastases	Very rare (<5%)	Uncommon (15–20%)	Very common (40–50%)	Common (35–40%)
Anatomic location of the tumour	Midline extending into cerebellopontine angle	Lateralized (cerebellar hemispheric) and superior	Midline and inferior (IV^th^ ventricle)	Midline and inferior (IV^th^ ventricle)
Proposed cell of origin	Lower rhombic lip progenitors	Granule neuron precursor cells in the external granular layer	Prominin+/CD133+ lineage neural stem cell	Premature glutamatergic neuronal networks
Candidate driver genes	CTNNB1, DDX3X	PTCH1, TP53, GLI2, KMT2D, MYC amplification	MYC amplification, SMARCA4, OTX2,	KDM6A, SNCAIP gain, MYCN amplification
Five-year overall survival	Excellent: >90–95%	Intermediate: 70% (50–85%)	Poor: 45% (30–60%)	Good: 75% (50–85%)

WNT = wingless; SHH = sonic hedgehog; LC/A = large cell/anaplastic; ND = nodular desmoplastic; MBEN = medulloblastoma with extensive nodularity.

**Table 2 diagnostics-14-00358-t002:** Patient, disease, and treatment characteristics of the study cohort (*N* = 67).

Characteristics	Number of Patients (%)
Median age (interquartile range) at diagnosis	12 years (9–18 years)
Gender	
Male	44 (65.7%)
Female	23 (34.3%)
Postoperative residual tumour (*n* = 57)	
<1.5 cm^2^	47 (82.4%)
≥1.5 cm^2^	10 (17.6%)
Metastatic status at diagnosis (*n* = 62)	
Nonmetastatic (M0)	57 (91.9%)
Metastatic disease (M+)	5 (8.1%)
Conventional risk stratification (*n* = 55)	
Average risk	39 (70.9%)
High risk	16 (29.1%)
Histological subtype	
Medulloblastoma (not otherwise specified)	22 (32.8%)
Classic	41 (61.2%)
Desmoplastic	3 (4.5%)
Large cell/Anaplastic	1 (1.5%)
Time interval from surgery to adjuvant radiotherapy (*n* = 40)	
≤6 weeks	17 (42.5%)
>6 weeks	23 (57.5%)
Craniospinal irradiation dose (*n* = 54)	
23.4–26 Gy ^$^	23 (42.6%)
35–36 Gy	31 (57.4%)
Craniospinal irradiation technique (*n* = 39)	
Conventional radiotherapy	1 (2.6%)
Three-dimensional conformal radiotherapy	16 (41.0%)
Intensity-modulated radiation therapy	22 (56.4%)
Adjuvant systemic chemotherapy (*n* = 54)	
Yes	39 (72.2%)
No	15 (27.8%)
Cumulative cyclophosphamide dose (*n* = 39)	
≤12 mg/m^2^	11 (28.2%)
>12 mg/m^2^	28 (71.8%)

^$^ One patient was planned for 23.4 Gy craniospinal irradiation but defaulted after 14.4 Gy/8 fractions.

**Table 3 diagnostics-14-00358-t003:** Patterns of relapse and salvage therapy in WNT pathway medulloblastoma experiencing an event (relapse/death) in the study (*n* = 9).

Sr No.	Age (Years) /Gender	Stage	CSI Dose at Initial Diagnosis	Pattern of First Failure	PFS	Salvage Therapy at Relapse	Final Outcome	OS
W1	9/Male	Nonmetastatic	36 Gy/18 fx	Tumour-bed recurrence	25 months	Re-RT and chemotherapy	Died of disease	80 months
W2	11/Male	Nonmetastatic	26 Gy/13 fx	Tumour-bed relapse plus metastases in brainstem, temporal lobe, and spine	37 months	Re-CSI (36 Gy/36fx) and chemotherapy	Died of disease	66 months
W3	13/Female	Nonmetastatic	35 Gy/21 fx	Leptomeningeal dissemination	25 months	Best supportive care	Died of disease	25 months
W4	10/Male	Nonmetastatic	14.4 Gy/8 fx (Incomplete RT)	Leptomeningeal dissemination	61 months	Best supportive care	Died of disease	62 months
W5	27/Male	Nonmetastatic	35 Gy/21 fx	Leptomeningeal dissemination	58 months	Best supportive care	Died of disease	60 months
W6	14/Male	Metastatic (frontal horn lesion)	35 Gy/21 fx	No evidence of disease progression/failure	15 months	Not applicable	Died of toxicity	15 months
W7	9/Male	Nonmetastatic	36 Gy/18 fx	Tumour-bed recurrence	56 months	Resurgery	Died of disease	60 months
W8	22/Male	Nonmetastatic	35 Gy/21 fx	Extra-neural metastases	83 months	Chemotherapy	Alive with disease	Not applicable
W9	15/Male	Nonmetastatic	Not known	Tumour-bed relapse plus metastases in frontal horn and multiple spinal metastases	59 months	Resurgery	Died of disease	67 months

WNT = wingless, CSI = craniospinal irradiation, RT = radiotherapy, PFS = progression-free survival, OS = overall survival, fx = fraction.

**Table 4 diagnostics-14-00358-t004:** Univariate analysis of survival outcomes for WNT-pathway medulloblastoma in the study cohort (*N* = 61).

Variables	Category	5-Year PFS (95%CI)	*p*-Value	5 Years OS (95%CI)	*p*-Value
Gender	Male	86.0% (73.8–100%)	0.480	92.6% (83.1–100%)	0.440
	Female	93.3% (81.5–100%)		93.7% (82.6–100%)	
Age at diagnosis	Child (≤16-years)	83.5% (83.0–100%)	0.722	89.5% (87.5–100%)	0.323
	Adult (>16-years)	90.0% (68.0–100%)		90.9% (72.0–100%)	
Residual disease	<1.5 cm^2^	84.5% (72.6–98.4%)	0.250	90.7% (80.9–100%)	0.260
	≥1.5 cm^2^	100% (NE)		100% (NE)	
Metastatic status	Nonmetastatic (M0)	89.1% (79.1–100%)	0.320	94.2% (86.4–100%)	0.270
	Metastatic (M+)	80.0% (51.6–100%)		80.0% (51.6–100%)	
Risk stratification	Average risk	93.3% (81.5–100%)	0.560	100% (NE)	0.640
	High risk	81.2% (63.9–100%)		87.9% (73.5–100%)	
Time interval (Surgery to RT)	≤42 days	87.3% (72.4–100%)	0.500	94.1% (83.6–100%)	0.440
	>42 days	89.4% (76.7–100%)		94.7% (85.2–100%)	
Dose of CSI	Low dose (14.4–26 Gy)	94.1% (83.0–100%)	0.441	100% (NE)	0.698
	High dose (35–40 Gy)	81.0% (68.1–97.0%)		84.6% (72.0–100%)	
Adjuvant chemotherapy	No	91.6% (77.2–100%)	0.990	100% (NE)	0.940
	Yes	84.3% (70.6–100%)		87.8% (75.2–100%)	
Cyclophosphamide dose	<12 gm/m^2^	92.8% (80.3–100%)	0.970	100% (NE)	0.970
	≥12 gm/m^2^	88.7% (77.4–100%)		92.6% (83.2–100%)	

WNT = wingless, PFS = progression-free survival, OS = overall survival, CI = confidence interval, RT = radiotherapy, CSI = craniospinal irradiation, NE = not estimable.

**Table 5 diagnostics-14-00358-t005:** Summary of outcomes, including patterns of relapse in WNT-MB patients treated adequately on molecularly informed prospective cohort studies and randomized controlled trials.

Trial Identity [Ref] and Registration	Risk Category	WNT-MB Patients	WNT-MB Failures	Patterns of Relapse			5-Year EFS/PFS	5-Year OS
				Local	Metastatic	Combined		
HIT 2000 [45] NCT00303810	Nonmetastatic (average risk)	15	0	0	0	0	100%	100%
SIOP PNET-4 [42] NCT01351870	Average risk	58	8	2	4	2	91%	95%
HIT 2000 [46] NCT00303810	Metastatic (high risk)	4	0	0	0	0	100%	100%
COG ACNS 0331 [11] NCT00085735	Average risk	64	4	4	0	0	93.3%	95.5%
SJMB-03 [12] NCT00085202	Average risk and high risk	46	0	0	0	0	100%	100%
COG ACNS 0332 [47] NCT00392327	High risk	14	1	0	0	1	92.9%	100%
SIOP PNET 5 HR+ [48] NCT00936156	High risk	3	0	0	0	0	100%	100%

WNT = wingless; MB = medulloblastoma; EFS = event-free survival; PFS = progression-free survival; OS = overall survival.

## Data Availability

All the data generated from this study are vested with the Principal Investigator and corresponding author. Data are not publicly available due to privacy issues but can be made available in an anonymized format upon reasonable request to the corresponding author.

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
