# Peer review of "Clinico-Radiological Outcomes in WNT-Subgroup Medulloblastoma"

_diagnostics, 2024, doi:10.3390/diagnostics14040358_

Round 1
Reviewer 1 Report
Comments and Suggestions for Authors
This is a well written paper. There are some duplications in the introduction.
One comment on the leptomeningeal spreading. This is not only by CSF but also hematogenic and possibly lymphatic. Maybe this could be a reason why it is in the frontal horns. Could you expand on this?
Another point to follow-up is on why 8 of the 9 patients who died in FU did get leptomeningeal spread, and the other ones not? Was there a difference in molecular biology? Myc of TP53 or even other ideas why this difference?
Finally, I miss a part of imaging. This is really relevant if we want to treatment some of the WNT patients less aggressive. Can we recognise WNT cases pre-operatively in nearly 100% of the cases, using MR features including MRS or not. Please expand.
Author Response
We thank the reviewers for their critical appraisal of our referenced manuscript providing constructive criticism and useful suggestions that has helped us to revise it meaningfully. Please find below our itemized response to the reviewers’ comments.
Reviewer #1
General comment: This is a well written paper. There are some duplications in the introduction.
Response to general comment: We are grateful to the reviewer for their appreciation that this is a well written paper. We have tried to reduce duplications in the introduction section in the revised manuscript based on the reviewer’s comments.
Specific comments:
Specific comment 1: One comment on the leptomeningeal spreading. This is not only by CSF but also hematogenic and possibly lymphatic. Maybe this could be a reason why it is in the frontal horns. Could you expand on this?
Reply to comment 1: We would like to re-emphasize that leptomeningeal dissemination (LMD) is the predominant pattern of metastases in medulloblastoma which occurs typically via the cerebrospinal fluid (CSF) pathways. Systemic or hematogenous spread is very rare in medulloblastoma seen in <1% of patient at initial diagnosis. Recently, the existence of circulating tumor cells (CTCs) in the peripheral blood has been demonstrated in patients with medulloblastoma that can explain hematogenous dissemination. Occasionally these CTCs can migrate towards the cranial and/or spinal leptomeninges leading to LMD. In the CNS, a lymphatic network has also been described, particularly in the meninges (within the dura mater), which facilitates CSF drainage, part of which in the subarachnoid space drains into the cervical lymph nodes connecting with the lymphatic circulation. This suggests that leptomeningeal metastasis occurs not only via the CSF, but can also occur through CNS lymphatics. As far as preferential spread of WNT-pathway medulloblastoma to the frontal horns is concerned, it is hypothesized that the ependymal lining of the ventricle has a unique microenvironment conducive to homing of WNT-medulloblastoma cells. All of this is now included in the revised submission.
Specific comment 2: Another point to follow-up is on why 8 of the 9 patients who died in FU did get leptomeningeal spread, and the other ones not? Was there a difference in molecular biology? Myc of TP53 or even other ideas why this difference?
Reply to comment 2: The more logical way to look at this is that WNT-MB patients who developed LMD succumbed to progressive disease, while patients who did not develop metastatic relapse had their disease under control becoming long-term survivors. Whether there was a difference in the molecular biology of WNT-MB in patients developing LMD versus those without metastatic relapse is an open question. Given that this was a retrospective dataset, we did not have information on other co-occurring molecular alterations (MYC amplification, TP53 mutation, gains of OTX2, etc) in WNT-MB that might have impacted upon survival. This has been acknowledged as a limitation of our study.
Specific comment 3: Finally, I miss a part of imaging. This is really relevant if we want to treatment some of the WNT patients less aggressive. Can we recognise WNT cases pre-operatively in nearly 100% of the cases, using MR features including MRS or not. Please expand.
Reply to comment 3: We thank the reviewer for this comment. Since imaging genomics was not a part of this retrospective study, we did not mention about radiomics/radiogenomics of medulloblastoma. Given the reviewer’s comment we have now included a small paragraph on radiomics/radiogenomics in the introduction section in the revised manuscript.
We sincerely hope that we have been able to address all the issues and concerns raised by both the reviewers in our revised submission making it acceptable for publication.
Reviewer 2 Report
Comments and Suggestions for Authors
The authors aim to determine the appropriate risk classification schemes and optimal treatment options for subgroup medulloblastoma (MB)- Wnt-MB, which has still not been defined. For this purpose, the authors conducted a retrospective study of clinical outcomes in patients with WNT-MB. This retrospective clinical study confirms that patients with WNT-MB who undergo maximal safe resection followed by appropriate radiation (chemotherapy) have excellent survival rates.
There are some minor concerns that the authors need to clarify or further address before the paper can be re-considered for publication:
Minor comment:
1. In Figures 1A and 1B: Kaplan–Meier curves of progression-free survival (A) and overall survival (B) Should compared with the control group. The resolution of the figure was too low.
2. In Figure 2, the author should provide three patient images respectively and add arrows to point out the locations described in the figure legend.
3. The conclusion is too short; the author should describe the difference between Wnt-MB and other subgroup MBs.
Comments on the Quality of English LanguageAlthough the general format of the paper is appropriate,the authors should seek the help of professional editors or native speakers to improve their English writing.
Author Response
We thank the reviewer for his/her critical appraisal of our referenced manuscript providing constructive criticism and useful suggestions that has helped us to revise it meaningfully. Please find below our itemized response to the reviewers’ comments.
Reviewer #2
General comment: The authors aim to determine the appropriate risk classification schemes and optimal treatment options for subgroup medulloblastoma (MB)- Wnt-MB, which has still not been defined. For this purpose, the authors conducted a retrospective study of clinical outcomes in patients with WNT-MB. This retrospective clinical study confirms that patients with WNT-MB who undergo maximal safe resection followed by appropriate radiation (chemotherapy) have excellent survival rates.
Response to general comment: We are grateful to the reviewer for their appreciative comments.
Minor comments: There are some minor concerns that the authors need to clarify or further address before the paper can be re-considered for publication:
Minor comment 1: In Figures 1A and 1B: Kaplan–Meier curves of progression-free survival (A) and overall survival (B) Should compared with the control group. The resolution of the figure was too low.
Response to comment 1: We would like to inform that this is a retrospective cohort study looking at the outcomes of WNT-pathway medulloblastoma, hence there is no control group for comparison. We have improved the resolution of the Figure 1A and 1B in the revised submission.
Minor comment 2: In Figure 2, the author should provide three patient images respectively and add arrows to point out the locations described in the figure legend.
Response to comment 2: We have added the arrows in the figure to point out the location of relapse as suggested by the reviewer.
Minor comment 3: The conclusion is too short; the author should describe the difference between Wnt-MB and other subgroup MBs.
Response to comment 3: Since the article referred to outcomes of WNT-subgroup only, we did not include differences between WNT and other subgroups. Based on the reviewer’s comments, we have now added a new table summarizing the differences in the clinico-biological characteristics of the 4 broad molecular subgroups of medulloblastoma in the introduction section in the revised submission. In addition, we have slightly expanded the conclusion based on the reviewer’s comments.
Comments on the Quality of English Language: Although the general format of the paper is appropriate, the authors should seek the help of professional editors or native speakers to improve their English writing.
Response to comment on English language: We think that the quality of language used in the manuscript is acceptable. Nevertheless, based on the reviewer’s suggestion we have further revised the language as appropriate with the help of native English speaker.
We sincerely hope that we have been able to address all the issues and concerns raised by both the reviewers in our revised submission making it acceptable for publication.